# Transcriptomics of Wet Skin Biopsies Predict Early Radiation-Induced Hematological Damage in a Mouse Model

**DOI:** 10.3390/genes13030538

**Published:** 2022-03-18

**Authors:** Abdulnaser Alkhalil, John Clifford, Stacy Ann Miller, Aarti Gautam, Marti Jett, Rasha Hammamieh, Lauren T. Moffatt, Jeffrey W. Shupp

**Affiliations:** 1Firefighters’ Burn and Surgical Research Laboratory, MedStar Health Research Institute, Washington, DC 20010, USA; lauren.t.moffatt@medstar.net (L.T.M.); jeffrey.w.shupp@medstar.net (J.W.S.); 2Pain and Sensory Trauma Care Research Team, US Army Institute of Surgical Research, Fort Sam Houston, San Antonio, TX 78234, USA; john.l.clifford11.civ@mail.mil; 3Medical Readiness Systems Biology Branch, Center for Military Psychiatry and Neuroscience Research, Walter Reed Army Institute of Research, Silver Spring, MD 20910, USA; stacyann.m.miller.ctr@mail.mil (S.A.M.); aarti.gautam.civ@mail.mil (A.G.); rash.hammamieh1.civ@mail.mil (R.H.); 4Walter Reed Army Institute of Research, Silver Spring, MD 20910, USA; marti.jett-tilton.civ@mail.mil; 5Department of Biochemistry and Molecular Biology, Georgetown University School of Medicine, Washington, DC 20010, USA; 6Department of Surgery, Georgetown University School of Medicine, Washington, DC 20010, USA; 7The Burn Center, Department of Surgery, MedStar Washington Hospital Center, Washington, DC 20010, USA

**Keywords:** radiation, blood, hematocytes, genomics

## Abstract

The lack of an easy and fast radiation-exposure testing method with a dosimetric ability complicates triage and treatment in response to a nuclear detonation, radioactive material release, or clandestine exposure. The potential of transcriptomics in radiation diagnosis and prognosis were assessed here using wet skin (blood/skin) biopsies obtained at hour 2 and days 4, 7, 21, and 28 from a mouse radiation model. Analysis of significantly differentially transcribed genes (SDTG; *p* ≤ 0.05 and FC ≥ 2) during the first post-exposure week identified the glycoprotein 6 (GP-VI) signaling, the dendritic cell maturation, and the intrinsic prothrombin activation pathways as the top modulated pathways with stable inactivation after lethal exposures (20 Gy) and intermittent activation after sublethal (1, 3, 6 Gy) exposure time points (TPs). Interestingly, these pathways were inactivated in the late TPs after sublethal exposure in concordance with a delayed deleterious effect. Modulated transcription of a variety of collagen types, laminin, and peptidase genes underlay the modulated functions of these hematologically important pathways. Several other SDTGs related to platelet and leukocyte development and functions were identified. These results outlined genetic determinants that were crucial to clinically documented radiation-induced hematological and skin damage with potential countermeasure applications.

## 1. Introduction

Exposure to ionizing radiation (IR) results in immediate, short-, and long-term injuries based on the intensity of the exposure [1,2]. The outcome of radiation injuries is exacerbated by a victim’s comorbidities or concomitant burns, wounds, and trauma that are frequently encountered in nuclear detonation [1,2,3,4]. Measuring the absorbed irradiation (IR) dose, a principal determinant of survival, is challenging [4,5,6] and complicated especially under mass casualty circumstances caused by an accidental or deliberate nuclear incident in urban or battlefield setups [7]. Most of the methods adopted for radiation diagnosis rely on biological, clinical, or physical symptoms and tend to be laborious, time-consuming, and do not enable pre-emptive therapeutic intervention [8]. Currently, there is no FDA-approved test enabling an accurate assessment of an absorbed radiation dose [5] or dosimetric measurement of IR exposure. Lack of such tests or tools delays patients’ evaluation, triage, and the identification of the best treatment strategy and outlining of ideal resources deployment [9,10,11,12] in a radiation-exposure event.

Realizing these challenges, the Biomedical Advanced Research and Development Authority (BARDA) is supporting the development of four products for the measurement of an absorbed radiation dose. One or more of these products is anticipated to aid in improving the outcome of care ensuing a nuclear or radiation occurrence [9]. The products are diverse in their approaches and include the following: (1) a protein-based assay that interrogates a panel of three plasma proteins, namely the salivary α amylase 1A (AMY1A), the Fms-related tyrosine kinase 3 ligand (FLT3L), and the monocyte chemotactic protein 1 (MCP1), which are modulated by radiation in a dose-dependent fashion; (2) an mRNA-based approach aimed at detecting panels of 12 or 15 radiation-sensitive mRNA molecules in blood using qRT-PCR; (3) ligation to contiguous amplified DNA fragments; (4) a cytokinesis-block micronucleus (CBMN) assay [13] aimed at detecting the dose-dependent induced micronuclei in lymphocytes [14,15,16]. Other methods with dosimetric potential are cell-based tracking radiation-induced hematocytopenia that follows the kinetics of lymphocytes, neutrophils, and platelet depletion [12,17,18]. Time to onset of clinical symptoms, such as time to emesis, was also proposed [19]. The current guidance of triage adopts a grading system that averages the injury grades (1–4) to the hematopoietic, the gastrointestinal (GI), the cutaneous (C), and neurovascular systems by following the time to onset of vomiting, diarrhea, abdominal pain, skin redness, rash, burns, and neurological symptoms to determine the level of damage to a patient’s health [20,21,22]. Patients with an average grade of 1 are candidates for ambulatory observation while those graded at 4 are candidates for palliative care. Patients average grade of 1–3 with hemopoietic system injury between 1–4 are subjects of treatment of the hematopoietic subsyndrome of acute radiation injuries. Treatments involve correcting the leukocytopenia and thrombocytopenia by transfusion of blood or specific hematocytes and stimulating synthesis of homogeneous blood cells using the traditional (filgrastim) or pegylated (pegfilgrastim) granulocyte colony-stimulating factor (G-CSF), or the granulocyte-macrophage colony-stimulating factor (GM-CSF) or sargramostim. Other platelet cytokines, such as romiplostim, are used to increase platelet production via binding and activation of the thrombopoietin (TPO) receptor [23].

Acute radiation syndrome (ARS) manifests as a compiled multi-tissue and multi-system injury with hematopoietic damage that is central to victim survival in the rescuable dose range (2–6 Gy) where correction of the hematocytopenia improves patients’ survivability. Therefore, early detection of radiation-induced damages to hematopoiesis and a better understanding of the underlying mechanisms postulate an important therapy intervention site for outcome mitigation. Genomic changes in blood samples irradiated in ex vivo [24,25,26,27,28,29,30,31] or in vivo [32,33,34] experiments were studied to address the immediate effects of radiation in blood cells using different animal models [32,33]. Most previous studies sought dosimetric biomarkers to enable radiation dose reconstruction and provide insights on ARS pathogenesis or mechanistic information about IR lethality, including damages to hematopoiesis and related immune responses [35], potential infection, or digestive and neurological system damages. While informative, most of these studies overlooked the essential role of the microenvironment [36] or extracellular matrix (ECM) [37,38,39] in hematopoiesis and the response to radiation in general. Results from studies using bone marrow samples in vivo were more translatable because they maintained a level of the native environment of progenitor cells, which is central to hematopoiesis. Similarly, the ECM-exchanged signals with resident leukocytes in the skin are key to these cells as well as ECM functions and the homeostasis of skin in general [35,40]. In addition to their important local role, skin-resident leukocytes exhibit systemic effects via the innate and adaptive immune systems [35,41] with their antimicrobial effects compromised in ARS and associated with increased death from infection [42,43]. Despite the different microenvironments of hematocytes and skin cells, a level of functional redundancy and responses through interaction with ECM can be distinguished [44], and collecting biopsies using non-absorbent conditions enables molecular assessment of a pool of the two cell types [45]. As skin and blood components are unshielded, as with bone marrow, it was proposed that they are more susceptible to radiation toxicity and immediate IR damages. Using a mouse radiation model, this work sought to investigate the potential of genomics in identifying radiation-induced injuries in the skin and peripheral blood with attention to the confounding associated logistics of the model [46]. Biopsies were collected at five time points distributed over 28 days after different IR doses. The potential of transcriptomics in the early identification of radiation-inflicted hematological injury and the determination of underlying transcriptomic alterations with corresponding biological functions were evaluated.

## 2. Material and Methods

### 2.1. Ethics

All animals in the study were handled according to facility standard operating procedures under the animal care and use program accredited by the Association for Assessment and Accreditation of Laboratory Animal Care International (AAALAC) and Animal Welfare Assurance through the Public Health Service (PHS). All performed animal work was reviewed and approved by the MedStar Health Research Institute’s Institutional Animal Care and Use Committee (IACUC-2010-021).

### 2.2. Animal Preparation and Radiation Treatment

Male C57BL/6 mice were purchased from The Jackson Laboratory (JAX, Bar Harbor, ME 04609, USA) at eight weeks old. The mice were acclimated for one week before initiating IR exposures and specimen collection. All animal work and sample collection were completed before animals reached 14 weeks of age. Animals were housed at a density of five animals per cage under standard housing conditions of food, temperature, water, and 12/12 h of light/dark cycles.

#### 2.2.1. Radiation Treatment

Mice in groups of five were placed in a round container split similarly to a pie into five equally-sized triangular compartments where in each house there was a mouse with its head pointing to the center of the container, which was covered with the same vented lid. The container was placed under a linear accelerator (Clinac 2100EX Manufacturer: Varian Medical, Crawley, UK) with its field size set at 32 cm × 32 cm to ensure coverage of the whole container. Based on the desired dose, monitor units (MU) were calculated and delivered half from the anterior and half from the posterior (standard AP/PA technique). The energy of the bean used to deliver the dose was 6 MV photons, which was run at a dose rate of 600 MU per minute. Machine output was calibrated following the TG51 protocol [47]. The mice received whole-body X-ray exposures (0, 1, 3, 6, or 20 Gy) while under anesthesia using IP injection of 300 μL ketamine (3 mg) and 50 μL of xylazine (3 mg) in saline. Isoflurane at concentrations of 2–5% was used in a controlled gas flow box or through a nose cone for the maintenance of anesthesia as needed.

#### 2.2.2. Sample Collection and Post-Irradiation Observation

Animals returned to the housing facility and skin biopsies were collected from each animal at day 0 (hour 2, h2), day 4 (d4), day 7 (d7), day 21 (d21), and day 28 (d28) post-irradiation. Briefly, the animals’ dorsa were shaved using standard veterinary clippers and a 1 cm^2^ biopsy was collected. Biopsy sites were closed using prolene sutures (Ethicon, Johnson & Johnson, NJ, USA). Animals exposed to 0, 1, 3, 6 Gy survived the full experiment time course and did not show signs of pain or distress after biopsy or during housing. Mice exposed to 20 Gy showed decreased activities by post-exposure day 6 and developed signs of distress, lethargy, and dehydration by day 7. Mice in this exposure group were euthanized on the same day when morbidity signs were observed per humane endpoint criteria defined in the IACUC-approved study protocol. Mice in the sham group were transported to the same radiation facility as the mice in the radiation groups, transferred to the radiation table, and returned to the housing facility with no radiation exposure. At designed time points, sham animals were anesthetized, shaved, and biopsied following the exact procedure of irradiated groups. At the end of the experiment time course, euthanasia was performed via exsanguination using cardiac puncture under anesthesia. Death was confirmed by lack of pedal and corneal reflexes and opening of the thoracic cavity to ensure lack of heartbeat.

### 2.3. Microarrays and Data Preparation

#### 2.3.1. RNA Extraction

Total RNA was isolated from liquid nitrogen flash-frozen biopsies after thorough grinding in a cold mortar and pestle. Each grounded biopsy was transferred into a 1.5 mL tube containing 1 mL TRIzol reagent (Invitrogen, Thermo Fisher, Waltham, MA, USA) and RNA was isolated following the manufacturer’s protocol. Concentrations and quality of yielded RNA were assessed using NanoDrop 1000 (Thermo Fisher, Waltham, MA, USA) and the Agilent 2200 Tapestation system (Agilent Technologies, Santa Clara, CA, USA). Isolated materials were aliquoted and stored at −80 °C until further use.

#### 2.3.2. Microarrays

25–200 nanograms of RNA was used following Agilent’s two-color array workflow utilizing the Two-Color Low Input Quick Amp Labeling Kit, Two-Color RNA Spike-In Kit, Gene Expression Hybridization Kit, and Gene Expression Wash Buffer Kit (Agilent Technologies, Santa Clara, CA, USA) following all manufacturer’s instructions. Briefly, samples and purchased reference RNA (Agilent Technologies) were reverse transcribed and labeled with Cy-5 and Cy-3 dyes respectively. All samples were then purified using the RNeasy Mini Kit (Qiagen, Hilden, Germany) and quantified on Nanodrop. Labeled cDNAs were simultaneously hybridized for 17 h at 65 °C on Agilent 4 × 44 K Whole Mouse Genome Microarray Kit (GPL7202: Agilent-014868) then slides were washed (Agilent Technologies, Inc., Santa Clara, CA, USA).

#### 2.3.3. Data Preparation and Analysis

Arrays were immediately scanned using an Agilent G2505C Scanner (Agilent Technologies Inc, Santa Clara, CA, USA). Images were processed using Agilent’s default Feature Extraction software v11.0.1.1 and analyzed using custom R scripts to obtain lists of probe sets differentially expressed. Minimum information about a microarray experiment (MIAME)-compliant intensity, quality, and normalized ratio data for this series of experiments have been deposited in the gene expression omnibus (GEO) database maintained by the National Center for Biotechnology Information (accession no. GSE185149). Uncentered Pearson clustering was done with tools developed by the Division of Computational Bioscience of the Center for Information Technology and the Cancer Genetics Branch of the National Human Genome Research Institute at the NIH. Fold change of gene expression was calculated by normalization of the transcriptomes at different doses of X-ray exposure over all time points to that of the sham-treated animals. Changes in gene expression at Benjamini–Hochberg FDR adjusted *p* < 0.05 were deemed significant. Further narrowing in the selection of significantly differentiated genes was performed on an excel sheet of all elements of the array by sorting based on the FC selection of elements where FC > abs 2 and *p*-value < 0.05 only. These lists were crossed with lists of annotated SDTG lists obtained after loading to ingenuity pathway analysis (QIAGEN Inc., https://www.qiagenbioinformatics.com/products/ingenuitypathway-analysis, accessed between 1 August 2021 and 15 December 2021). Only genes that were common to both lists were included in all subsequent analyses. Top pathways were reported based on abs z scores from IPA. Lists of genes in the reported pathways were obtained from IPA with no further processing of additional cutoffs. Analyses were performed comparing results of significantly differentially regulated genes after exposure to different doses of X-rays over several time points.

## 3. Results

Five groups of mice each consisting of five animals were used in the study. Each mouse group received 1, 3, 6, or 20 Gy of whole-body X-ray radiation and the fifth group followed the exact steps of exposure protocol without radiation for use as a reference. Skin biopsies were collected from each mouse at h2, d4, d7, d21, and d28 post-exposure. Transcriptomes in each biopsy were examined using microarrays. Mice exposed to 20 Gy did not complete the study time course and were euthanized by d7 following the humane endpoint defined by the IACUC-approved study protocol underscoring the lethality of the 20 Gy IR dose (Figure 1). All animals in the reference group and groups that received low IR doses of 1, 3, 6 Gy survived the study time course, indicating that these doses were in the sublethal range.

Detailed analysis of the significantly differentially transcribed genes (SDTGs) after irradiation was described in a separate paper (in review) and showed a larger number of predominantly downregulated SDTGs after exposure to a lethal dose (20 Gy). This trend was reversed after exposure to sublethal doses (1, 3, or 6 Gy) where the numbers of SDTGs were less and were primarily upregulated. To examine whether transcriptomics could identify genetic modulations underlying the clinically reported hematological alterations induced by radiation and pathway enrichment analysis of the SDTGs (*p*-value ≤ 0.05 and FC ≥ 2) after lethal exposure was performed. The initial analysis included data from biopsies collected at h2, d4, and d7. Filtering the identified pathway using stringent statistical cutoffs (Abs z-score value ≥ 2 and −log *p* ≥ 1.3) identified three pathways that played a critical role in hematopoiesis and hematological response among the top of functionally altered pathways. The glycoprotein 6 (GP-VI) signaling pathway, the dendritic cell maturation pathway, and the intrinsic prothrombin activation pathway were at the top of the list of affected pathways (Figure 2). All three pathways at all three time points (TPs) were predicted to be strongly inactivated (Figure 2) after lethal IR dose exposure. Investigation of the statuses of these and other pathways after exposure to sublethal doses during the same first post-exposure week, using the same statistical cutoffs for the SDTGs and pathways, showed that the intrinsic prothrombin activation pathway and the GP6 were still among the top affected, and whenever a pathway was significantly identified after sublethal IR doses it showed an activity prediction that contrasted that predicted after lethal exposure. The immunologically and hematopoietically important interleukine (IL6)-signaling pathway was identified to be significantly activated at several TPs after exposure to sublethal doses but not the lethal dose (Figure 3).

Further analysis that included the late time points (d21, d28) after sublethal IR dose exposure showed that all three pathways that play important roles in hematological responses were among the top five identified pathways in support of transcriptomics application in detecting radiation-induced hematological injury (Figure 4). While these pathways showed inversed activity statuses during the first week between lethal and sublethal doses, the intrinsic prothrombin activation, and the GP6 signaling pathways were inactivated in the late time points after sublethal exposures mimicking the same activation status after lethal exposure, which suggested a late deleterious phase of response to sublethal doses in concordance with clinical observations at these levels of IR.

The GP6 signaling and the prothrombin signaling pathways contain many common genes. Significant modulation in the transcription of several collagens, laminin β 1, and calmodulin-like 5 genes introduced the GP6 signaling pathway as a main modulated pathway by IR (Table 1). Similarly, many of these collagen genes, in addition to kallikrein-related peptide 7 and 8 genes, were the main factors in the identification of the intrinsic prothrombin activation pathway (Table 1). Many of these genes were upregulated after sublethal IR doses exposure during the first post-exposure week then they turned to downregulation in the later TPs to mimic their regulation status after a lethal IR exposure.

Transcription modulations in many genes encoding interleukins and heat shock proteins among other genes (Figure 5) introduced the IL-6 signaling pathway as a pathway that was majorly affected by radiation. This pathway shared only the COL1A1 with the GP6 signaling and the intrinsic prothrombin activation pathways.

Several other genes that play an important part in hematological response and hematopathology were identified among the top differentially transcribed genes. The platelet-derived growth factor-like platelet factor 4, the leukocyte immunoglobulin-like receptor B3 (*LILRB3*), the heme binding protein 2 (*HEBP2*), the hemicentin 2 (*HMCN2*), the hemoglobin subunit β (*HBB*), and the hemoglobin subunit α 1 and 2 (*HBA1/HBA2*) genes were all downregulated sharply at all TPs after exposure to a lethal dose. Many of these genes were upregulated at different TPs after exposure to sublethal doses (Table 2).

## 4. Discussion

Hematological perturbation after radiation exposure has been documented in humans and other species [48,49,50]. A damage-compensating enhanced proliferation of stem cells dominated responses at low IR doses (≈ 0.2–0.3 Gy) to replenish irreparably affected hematocytes. These responses were overwhelmed or defective at IR dose exposures > 0.5 Gy and a steady decrease in the counts of lymphocytes and thrombocytes was noted and exacerbated by increased IR dose. Doses > 4.5 Gy or cumulative doses > 8 Gy induce red bone marrow (RBM) hypoplasia with inhibition of all blood cell lineages. The high levels of radiation dose exposures were associated with a poor prognosis and were generally lethal. In agreement with these clinical observations, recent work from our lab established a correlation between the RNA integrity number (RIN) retrieved from blood samples and radiation dose in mice [50]. Late time points (>7 days) after exposure to high doses (>5 Gy) yielded lower RNA quantity and quality in concordance with the reported leukocytopenia and recovered only in lower dose exposures. Results of the current study showed that mice that received 20 Gy did not survive past day 7 and transcription in skin biopsies from these mice was stalled with a predominant downregulation of a large number of genes as early as h2 after irradiation. Animals that were exposed to sublethal doses (1, 3, 6 Gy) survived the full-time course of the study (28 days). Transcriptional changes in biopsies after sublethal exposures involved smaller numbers of SDTGs relative to that after lethal exposure. The affected SDTGs after sublethal exposures were mainly upregulated during the first post-exposure week, many of which shifted to a downregulation state at later TPs, mimicking responses observed after a lethal dose. The proportion of genes that reverted to downregulation during the later TPs increased with increasing IR doses within the sublethal IR range. This longitudinal regulation pattern suggested a potential late deleterious effect for high sublethal doses (i.e., 6 Gy). The impact of these transcriptional modulations on animal health was beyond the study time course, however, they aligned with the deleterious long-term effects of radiation in the stochastic sublethal range [51].

The early divergence in transcription responses in lethal and sublethal IR doses provided an excellent tool for detecting the intensity of the exposure, guiding triage, and predicting outcomes. For example, no changes in thrombocyte and neutrophil counts were found during the first post-exposure week and it took 36 and 31 days after a 1 or 3 Gy dose, respectively, to reach the lowest cell counts [18]; however, results in this report identified many SDTGs that could predict thrombocytes and coagulation functional abnormalities via modulations of the GP6 signaling and the intrinsic prothrombin activation pathways as early as h2 after exposure. The activation of coagulation signaling at sublethal doses (1–6 Gy) and inactivation after a lethal IR dose were consistent with the correlation of the increase of coagulation instability and increased doses of IR exposure.

Modulations in several types of collagens were a major identifier of the GP6 signaling and the intrinsic prothrombin signaling pathways. The disrupted transcription of different types of collagens identified here was congruent with similar modulations in collagen synthesis in patients treated with radiation for head and neck cancer [52] or from in vitro experiments using irradiated fibroblasts [53]. The downregulation of collagens has a multitude of implications on cell biology. Some of these implications are direct, as seen in the reported changes in skin structure and functions after radiation [35,52,53,54,55], or in the increased bleeding [56,57] upon changes in collagen–platelet interactions, which trigger the activation of platelets and initiate the coagulation cascade. While these interactions were likely to contribute to the mid or late effects of radiation as the transcriptional changes are translated into a modified extracellular matrix structure or composition, immediate impacts of collagen transcription modulations mediated by the intracellular processes associated with or secondary to collagen translation, post-translational modifications, and intracellular assembly and transport occur much earlier. These effects were not limited to skin cells but include leukocytes because they express collagens under different conditions, including wound healing and in response to stress [58].

A reduction in the secretory vesicles that transport the procollagen to extracellular locations due to downregulation in several types of collagen would affect the interlinked process of transport of the calcium-loaded matrix vesicles [59]. As such, decreases in collagen synthesis would indirectly alter calcium deposition, which postulates a possible explanation to the reported susceptibility of cancer patients to bone fractures after radiation therapy [60,61,62,63,64]. Accumulation of calcium-loaded matrix vesicles and possible non-specific release in reduced collagen secretory vesicle conditions would also indirectly affect the homeostasis of coagulation frequently seen after radiation exposure [65]. The important role of the different types of cell vesicles in the delivery of a wide variety of cell cargo, including collagen, to the extracellular matrix and neighboring cells, would affect hematopoiesis and coagulation, among other functions. Identification of the actin cytoskeleton signaling pathway among the top affected pathways during the first post-exposure week supports alterations in secretory vesicles by radiation. Similarly, the secretory vesicles are known to transport Wnt proteins [66], which regulate calcium inside the cell via regulation of the noncanonical Wnt/calcium pathway. Although this pathway was identified as modulated and did not meet the significance threshold adopted here, several findings such as the identification of calcium signaling, actin cytoskeleton signaling, and downregulation of calmodulin-like 5 (CALML5) at the lethal IR dose and late TPs of sublethal doses point indirectly to a modulation of the Wnt/Ca pathway by radiation, which result in hematopoiesis and hematological alterations [67].

Collagens expression was implicated in vasculature damage [68]. Depending on the exposure intensity, injuries to vasculature ranged from reduced lumen diameter to ample macro- or microvessels loss. At low doses, the damage was lessened by the activation of detoxification mechanisms, repair and turnover of damaged molecules, or through an increase in the cell proliferation that replaces irreparable or apoptotic cells. High doses were associated with irreversible damages [69]. Alterations in collagen synthesis by dysfunctional mesenchymal stem cells (MSCs) were linked to T cell large granular lymphocyte leukemia pathophysiology [70] and many preclinical animal studies showed that MSCs can enhance regeneration of tissues and accelerate angiogenesis and re-epithelialization [71], potentially via synthesis of extracellular vesicles [72]. Interestingly, the advantages of using MSCs in the management of accidentally IR exposed individuals were attributed to its secretory activity, primarily the anti-inflammatory effects of the extracellular vesicles (EVs) [73]. This was supported by the alleviatory effects of none-irradiated mouse serum to radiation-induced hematopoietic system damages via improvements of the systemic environment and exosomal functions [74]. Other reports have linked the decrease in the synthesis of different collagen types to hematopoiesis and Th1/Th2 responses [75,76,77,78,79], which supports a correlation of the variety of modulated collagen transcription in this work and the hematological changes following radiation exposure. The multiple observations correlating modulations of collagens to cell survival and other important organ functions, including hematological systems, before deposition in the extracellular matrix were intriguing and need a further investigation. Indications of vesicle involvement and cotransport of important cargo, in addition to collagens, pose a potential mechanism for future studies.

The downregulation of serin protease kallikrein (3, 5-8)-related peptidases 3,5-8 (*KLK 3, 5-8*) after a lethal dose and in several later TPs after sublethal IR doses would translate to a disruption in the coagulation signaling and contribute to coagulopathies and other cutaneous symptoms reported in affected patients [80,81]. These genes were upregulated at all the TPs when significantly identified during the first week after sublethal exposure, suggesting survival advantages in addition to their wide applications as a marker for several diseases [81].

Several other genes that play important roles in the function and development of erythrocytes were identified. The erythroblast membrane-associated protein (*ERMAP*) and the erythroid differentiation regulator (*Erdr1*) were among the few upregulated genes in response to lethal IR exposure. These two genes encode proteins that modulate erythrocyte adhesion properties [82] and negatively regulate cell migration and regulation [83], respectively. Other erythrocyte-related genes such as the hemoglobin subunit α 1 and 2 (*HBA1* and *2*), hemoglobin subunit β (*HBB*), hemopexin (*HPX*), heme binding protein 2 (*HEBP2*), and hemojuvelin BMP co-receptor (*HJV*) were downregulated at all TPs after lethal IR exposure and sporadically at TPs of sublethal exposures including early TPs. These genes encode subunits of hemoglobin and other accessory proteins that play important roles in blood coagulation, platelet aggregation, nitric oxide transport, oxygen transport, peroxide catabolism, hemopoiesis, glutathione metabolism, bicarbonate transport, the collapse of mitochondria membrane potential, calcium transport, positive regulation of transcription from RNA polymerase II promoter, and iron ion homeostasis. The differential transcription of these sets of genes and corresponding functional damages provide new insights into the clinically observed dysfunctions after irradiation [84]. Other SDTGs important for the functions of platelets and leukocytes, including the secretory leukocytes peptidase inhibitor (*SLP1*) and plasminogen (*PLG*), were upregulated after lethal IR dose and late TPs (d21, d28) after large sublethal doses. The PLG gene encodes an inactive form of a serine protease that cleaves many extracellular matrices and blood clot components [85] while the SLP1 encodes a serine proteases inhibitor with antimicrobial properties that protects epithelial surfaces as a part of innate immune responses [86]. The leukocyte immunoglobulin-like receptor B3 (*LILRB3*) and the platelet-derived growth factor-like (*PDGFL*) were downregulated after the lethal IR dose and upregulated at d4 after 6 Gy exposure. The LILRB3 protein is thought to ameliorate inflammatory responses and limit autoreactivity by binding to MHC class I molecules on antigen-presenting cells and might confer a survival advantage when upregulated, and the PDGFL protein has tumor suppression properties. Correcting hematological damages after sublethal IR exposures are possible and are likely to improve survival rates, however, lethal IR doses resulting in immediate and long-term damages to different tissues, including the substantial bone marrow suppression that prevents recovery, is still a challenging task because they are associated with the long-lived free radicals and reactive oxygen species with an abundance of pro-inflammatory cytokines/chemokines [54] and often exacerbated with burns, wounds, or trauma from blasts injuries.

In addition to the use of granulocyte–colony stimulating factor (G-CSF), keratinocyte chemoattractant (KC), and controlling hematological damages after sublethal doses to improve survival after exposure to sublethal range doses [87], targeting the hematological instability through correction of the transcription of genes underlying the abnormalities in GP6 signaling, intrinsic prothrombin signaling pathways, and microenvironment of hematopoiesis might improve outcomes after IR exposure. Extending the findings of this work to use in radiation countermeasures requires additional investigations using human specimens.

## Figures and Tables

**Figure 1 genes-13-00538-f001:**
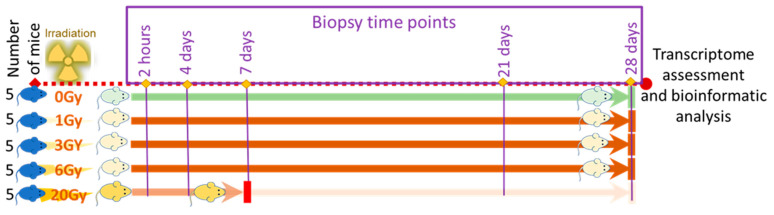
Study design.

**Figure 2 genes-13-00538-f002:**
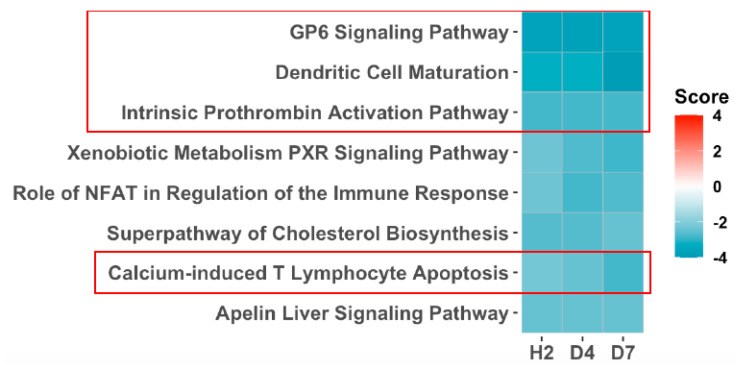
Pathway enrichment analysis of SDTGs (*p* ≤ 0.05 and FC ≥ 2) after exposure to the lethal dose (20 Gy). Pathways that did not pass filtration criteria (z-scores ≥ Abs 2 and −log *p* ≥ 1.3) at a time point are dotted.

**Figure 3 genes-13-00538-f003:**
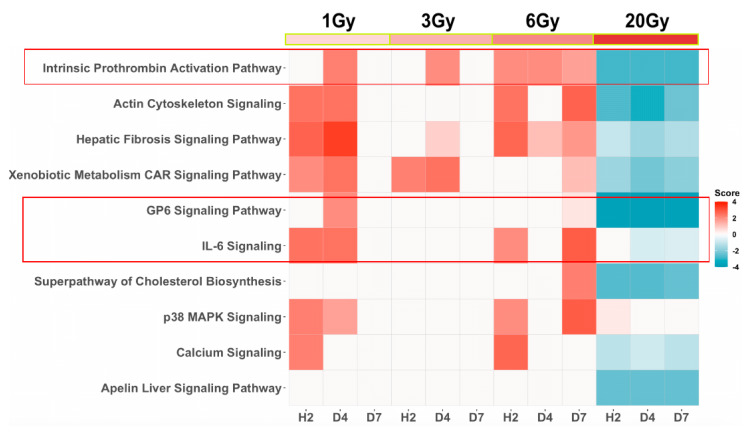
Pathway enrichment analysis of SDTGs (*p* ≤ 0.05 and FC ≥ 2) in biopsies collected during the first week after exposure to lethal and sublethal IR doses.

**Figure 4 genes-13-00538-f004:**
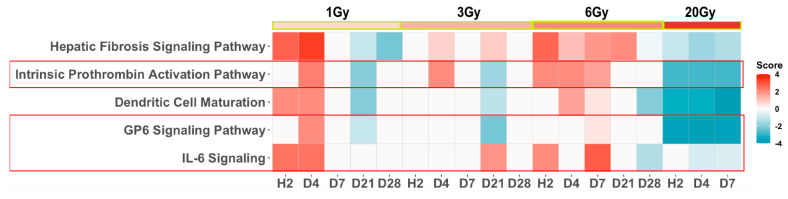
Pathway enrichment analysis of SDTGs (*p* ≤ 0.05 and FC ≥ 2) from all TPs and IR doses (pathways z-scores > Abs 2.75 and −log *p* > 4 or *p* <0.0001). Pathways in red boxes contribute to hematological homeostasis.

**Figure 5 genes-13-00538-f005:**
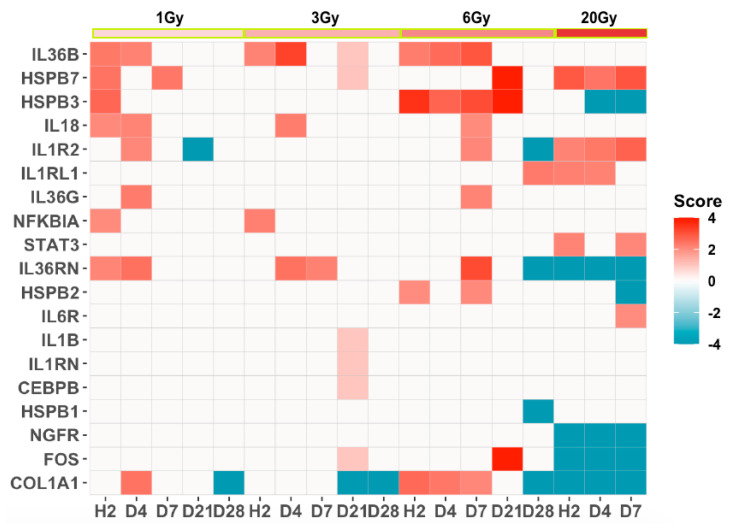
Heatmap of the SDTGs (FC > 2 and −log *p* > 1.3) enriched to the IL-6 signaling pathway.

**Table 1 genes-13-00538-t001:** Significantly differentially transcribed genes underlying the identification of GP6 signaling and intrinsic prothrombin signaling pathways. These two pathways share several key genes (*p* ≤ 0.05 and FC ≥ 2). Red-filled cells indicate upregulation and green-filled cells indicate down-regulation.

GP6 Signaling	Intrinsic Prothrombin	Symbol	Entrez Gene Name	1 Gy					3 Gy					6 Gy					20 Gy		
				h2	d4	d7	d21	d28	h2	d4	d7	d21	d28	h2	d4	d7	d21	d28	h2	d4	d7
×		LAMB1	laminin subunit bate 1																2.078		2.041
×	×	COL1A1	collagen type I alpha 1 chain		2.44			−2.485				−2.467	−2.007	2.612	2.378	2.1		−2.314	−24.32	−23.33	−35.28
×	×	COL1A2	collagen type I alpha 2 chain				−2.063					−2.935							−22.21	−20.32	−31.16
×	×	COL2A1	collagen type II alpha 1 chain									−2.093		2.306	2.587				−3.441	−2.971	−4.35
×	×	COL3A1	collagen type III alpha 1 chain				−2.081					−3.277							−31.75	−26.58	−47.74
×		COL4A4	collagen type IV alpha 1 chain																		−2.175
×		COL5A1	collagen type V alpha 1 chain																−4.282	−3.514	−5.265
×		COL5A2	collagen type V alpha 2 chain																−2.437		−2.762
×	×	COL5A3	collagen type V alpha 3 chain													−2.03			−3.209	−3.717	−4.282
×		COL6A1	collagen type VI alpha 1 chain																		−2.053
×		COL6A2	collagen type VI alpha 2 chain																−2.071	−2.024	−2.409
×		COL6A3	collagen type VI alpha 3 chain																−4.002	−3.526	−5.218
×		COL7A1	collagen type VII alpha 1 chain		2.143														−2.741	−3.039	−3.194
×		COL15A1	collagen type XV alpha 1 chain				−2.051					−2.231							−6.312	−5.899	−7.502
×		COL16A1	collagen type XVI alpha 1 chain																−2.501	−2.24	−2.314
×	×	COL18A1	collagen type XVIII alpha 1 chain																−2.287	−2.378	−2.218
	×	KLK7	kallikrein related peptidase 7	2.427	2.629			−2.687		2.092	2.015					3.021		−3.516	−2.416	−2.591	−2.46
	×	KLK8	kallikrein related peptidase 8	2.6	2.544				2.194	2.348	2.026					2.166			−2.366	−2.534	−2.983
×		CALML5	calmodulin like 5	2.045	2.438			−2.683								2.995		−3.383	−3.302	−3.33	−2.989

**Table 2 genes-13-00538-t002:** Significantly differentially transcribed genes (SDTGs, *p* ≤ 0.05 and FC ≥ 2) with an impact on hematological functions. Red-filled cells indicate upregulation and green-filled cells indicate down-regulation.

Gene Name	1 Gy					3 Gy					6 Gy					20 Gy			Location
	h2	d4	d7	d21	d28	h2	d4	d7	d21	d28	h2	d4	d7	d21	d28	h2	d4	d7	
*Erythroblast membrane associated protein (ERMAP)*															2.86	2.88	3.07		Cytoplasm
*Erythroid differentiation regulator 1 (EErdr1)*															2.40		2.32		Other
*HBA1/HBA2 hemoglobin subunit α 1 and 2*	2.00											−2.18			−2.72	−3.70	−6.39	−4.40	Extracellular space
*Hemoglobin subunit β (HBB)*	3.15	2.31					−2.12	2.35				−2.83			−3.01	−3.41	−6.06	−4.87	Cytoplasm
*Hemopexin (HPX)*										2.22									Extracellular space
*Hemicentin 2 (HMCN2)*													2.41			−2.79	−2.70	−2.73	Cytoplasm
*Heme binding protein 2 (HEBP2)*	2.19	2.26	2.03	2.04							2.78	2.54	2.31	3.46		−2.15	−2.06	−2.33	Extracellular space
*Hemojuvelin BMP coreceptor (HJV)*																	−2.40		Plasma Membrane
*Secretory leukocyte peptidase inhibitor (SLP1)*									2.01						2.19	4.73	4.00	3.99	Cytoplasm
*Plasminogen (PLG)*										2.26						2.28	2.36	2.51	Extracellular space
*Leukocyte immunoglobulin like receptor B3 (LILRB3)*												2.01				−2.90	−2.36	−2.85	Plasma Membrane
*Platelet factor 4*												2.62						−2.36	
*Platelet derived growth factor like*									−2.53			2.03				−3.14	−2.33	−3.67	Plasma Membrane

## Data Availability

Minimum information about a microarray experiment (MIAME)-compliant intensity, quality, and normalized ratio data for this series of experiments have been deposited in the gene expression omnibus (GEO) database maintained by the National Center for Biotechnology Information (accession No. GSE185149).

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
