# Peer review of "Transcriptomics of Wet Skin Biopsies Predict Early Radiation-Induced Hematological Damage in a Mouse Model"

_genes, 2022, doi:10.3390/genes13030538_

Round 1
Reviewer 1 Report
Microarray data should be open to reviewers.
The authors write that they are registered with the Gene Expression Omnibus (GEO) database maintained by the National Center for Biotechnology Information (accession no. GSE185149).
However, I cannot see the data during the peer review period.
This makes it impossible for me to do peer review.
Inappropriate use of abbreviations.
Abbreviations are not explained in the ”TPs” abstract.
The abbreviation appears for the first time on page 2, line 46 of the “ECM”, but the explanation is given on page 2, line 49.
Checks are required for all abbreviations.
The format of the reference is inappropriate. In the papers of multiple authors, only the first author's name is written.
Author Response
We thank the reviewer for the valuable time invested in the critical review of the manuscript. We have addressed all the points and concerns brought up primarily by adjusting the manuscript. We hope that the reviewer will find our answers and adjustments satisfactory, and we are ready to answer any additional questions or inquiries he may still have. Below is our point-by-point response to the reviewers’ concerns.
Comments and Suggestions for Authors
Q1- Microarray data should be open to reviewers.
The authors write that they are registered with the Gene Expression Omnibus (GEO) database maintained by the National Center for Biotechnology Information (accession no. GSE185149).
However, I cannot see the data during the peer review period.
This makes it impossible for me to do peer review.
A1- The data will be open to the public in the next 6 months after data analysis of other aspects of the data is completed. We have arranged for the reviewer to access the microarray data. Please see below:
The following secure token has been created to allow review of record GSE185149 while it remains in private status:
mhcbyccshxcvfab
Please note the following points:
- This token allows anonymous, read-only access to GSE185149 and associated accessions while they are private;
- Treat the token as you would a password and realize that the token provides access to GSE185149 to anyone who uses it - we recommend you do not include the token anywhere except in a secure email to journal editors;
- If you want to revoke this access token, you can click the 'Reviewer access' button on your record which exposes the option to revoke this token.
Send the following information to journal editors who will circulate to reviewers requiring access to your private data:
- ----------
- To review GEO accession GSE185149:
- Go to https://urldefense.com/v3/__https://www.ncbi.nlm.nih.gov/geo/query/acc.cgi?acc=GSE185149__;!!D7IIWT94AA!vGalOhagO12tixJlggeHJ_G-lP12rGFq4NLlGRvTgINJTvQKYQYus9ofg8M25hR6Uko87D7w$
- Enter token mhcbyccshxcvfab into the box
- ----------
Q2- Inappropriate use of abbreviations.
Abbreviations are not explained in the ”TPs” abstract.
The abbreviation appears for the first time on page 2, line 46 of the “ECM”, but the explanation is given on page 2, line 49.
Checks are required for all abbreviations.
A2- Corrective steps are Done. Please see edits in the resubmitted version of the manuscript. Abbreviations are spelled out at the first time introduced in the manuscript.
Q3- The format of the reference is inappropriate. In the papers of multiple authors, only the first author's name is written.
A3- Done.
Reviewer 2 Report
1) Alkhalil et.al have done excellent work on transcriptomics to predict early radiation-induced hematological damages in mice. But they have not validated their transcriptomic data. They can validate certain proteins from major pathways at different timepoints to prove their observation. Transcriptomic data may or may not match their observations, so it is important to validate it.
2) Authors have not discussed the utility, application and limitation of their work in discussion. It would be interesting to see correlation of these pathways to human samples if data is available.
3) The 1st paragraph of result section can be included in method section as it is describing the study design.
4) The last part of 2nd paragraph of introduction is not properly referenced. Authors have used too many references in this articles which should be reduced.
5) Certain abbreviation are not expanded at their first incidence eg TP in abstract and ECM
Author Response
We thank the reviewer for the valuable time invested in the critical review of the manuscript. We have addressed all the points and concerns brought up primarily by adjusting the manuscript. In the rare case when the reviewer's point was not answered directly, we explained and reasoned our standpoint in a common and scientifically acceptable manner. We hope that the reviewer will find our answers and adjustments satisfactory, and we are ready to answer any additional questions or inquiries he may still have. Below is our point-by-point response to the reviewers’ concerns.
1) Alkhalil et.al have done excellent work on transcriptomics to predict early radiation-induced hematological damages in mice. But they have not validated their transcriptomic data. They can validate certain proteins from major pathways at different timepoints to prove their observation. Transcriptomic data may or may not match their observations, so it is important to validate it.
- We thank the reviewer for bringing up this important and valid point. We have contemplated into this point and we always verified and validated our transcriptomic data using RT PCR or protein-based assays, however, in this specific report the verification was obviated by the following criteria met by the data:
i) Microarrays of skin biopsies from different mice, at different time points, using different RNA extraction sessions, after different doses of IR exposures introduced a similar list of genes that were similarly differentially regulated. The statistics for this to occur on random ranks at significant p values (i.e., the data is self-assertive)
ii) Dysregulations of the indicated genes/pathways result in hematological homeostasis disruptions based on confirmed gene functions. This was concordant with known clinical manifestation and documented symptoms after radiation exposure (i.e., repair and recovery after low IR exposure and succumb to hemopathies and cutaneous disorders after large IR dose exposure).
iii) Some of the identified dysregulated genes in this work were reported by others after IR exposure. Our work is an extension and expansion of current knowledge.
2) Authors have not discussed the utility, application and limitation of their work in discussion. It would be interesting to see correlation of these pathways to human samples if data is available.
- We understand and appreciate the reviewer’s point. Changes to the hematological and cutaneous systems in patients undergoing radiation therapy or victims of accidental radiation exposure are well-documented and in the alignment of our findings. Changes to the identified pathways in this work will certainly result in hematological and cutaneous systems disruptions. We have added the following sentence to address the reviewer's point “Extending findings of this work to use in radiation countermeasures requires additional investigations using human specimens“.
3) The 1st paragraph of result section can be included in method section as it is describing the study design.
- We agree, however, because the paragraph and related figure provide insights into the survival in this specific animal model, we think the RESULTS section is a better fit.
4) The last part of 2nd paragraph of introduction is not properly referenced. Authors have used too many references in this articles which should be reduced.
- Done. New Citation inserted for the indicated paragraph. When possible, multiple references for the same point were reduced to decrease the total number of references in the manuscript.
5) Certain abbreviation are not expanded at their first incidence eg TP in abstract and ECM
- Done. Corrections were made in the manuscript. For reference TP: time point, ECM: extracellular matrix.
Round 2
Reviewer 1 Report
The authors were used two-color array methods.
It is known that the distribution of fluorescence intensity differs between Cy3 and Cy5, and the results may be biased depending on which combination is used for the experiment. Therefore, it is necessary to confirm the relationship between the array number and the sample conditions.
However, looking at GSE185149, I did not understand the relationship between the array and sample conditions.
Please describe which array performed the experiment under which condition in the manuscript, or describe it in the GSE185149 of GEO.
The n numbers of each group didn't described(2h, 4,7,21,28 days after 0,2,4,6 Gy, so 20 Group, 20 Gy 2h, 4,7 days later, 3Group, total 23Group?). Please describe them.
The authors extracted the gene of p ≤ 0.05, FC ≥ 2 as SDTGs and further analyzed it by pathway, but only the result is shown.
Please describe what kind of statistical test was performed Between which groups in order to extract SDTGs. Also, as a result, please provide information on the number of genes that are up-regulated or down-regulated of each group respectively.
Data on body weight change after X-ry irradiation is necessary to infer the condition of the mouse.
The reason for gene induction or suppression is the direct effect of radiation or the indirect effect.
Please clarify the pathway that was only partially changed is really induced or suppressed by radiation.
For example, from Figures 2 and 3, the GP6 signaling pathway remains unchanged at 3 Gy. Why doesn't it change with 3 Gy?
Also, the Apelin Liver Signaling Pathway changes only at 20 Gy, and it is not possible to determine whether it is due to radiation or just suppressed because it is in a lethal state.
Author Response
Reviewer 1, Round 2:
It is known that the distribution of fluorescence intensity differs between Cy3 and Cy5, and the results may be biased depending on which combination is used for the experiment. Therefore, it is necessary to confirm the relationship between the array number and the sample conditions.
However, looking at GSE185149, I did not understand the relationship between the array and sample conditions.
Please describe which array performed the experiment under which condition in the manuscript, or describe it in the GSE185149 of GEO.
A1- We followed the manufacturer's two-color microarray-based gene expression analysis protocol (version 6.9.1). Analysis for this protocol and the use of internal RNA control samples spiked into the tested samples should guard against any biases. Briefly, samples and purchased reference RNA (Agilent Technologies) were reverse transcribed and labeled with Cy-5 and Cy-3 dyes respectively. Purchased reference RNA was used as a common reference design method to control for the dye effect. All samples are labeled with Cy5, and all reference samples are labeled with Cy3 using Agilent’s two-color low Input Quick Amp Labeling, along with RNA spike-ins from Agilent's two-color RNA spike-in kit (Agilent Technologies, Santa Clara, CA) following the manufacturer’s two-color microarray-based gene expression analysis protocol (version 6.9.1).
The n numbers of each group didn't described(2h, 4,7,21,28 days after 0,2,4,6 Gy, so 20 Group, 20 Gy 2h, 4,7 days later, 3Group, total 23Group?). Please describe them.
A2- The experiment design included five groups of mice each consisting of 5 animals (25 mice total). A group out of the five was exposed to 0, 1, 3, 6, or 20Gy. Biopsies were collected sequentially at 2h, days 4, 7, 21, and 28. Animals exposed to 20Gy didn’t survive past day 7, hence, only 2h, day 4, and day 7 were available from this particular group. Please see Figure 1 for a schematic presentation of the design.
The authors extracted the gene of p ≤ 0.05, FC ≥ 2 as SDTGs and further analyzed it by pathway, but only the result is shown.
A3- Lists of the genes were filtered at p ≤ 0.05 & FC ≥ 2 before IPA, as indicated. Pathways analysis was performed and all results are presented in Figures 2, 3, 4 after passing z-scores and p-value threshold indicated in the legend of each figure.
Please describe what kind of statistical test was performed Between which groups in order to extract SDTGs. Also, as a result, please provide information on the number of genes that are up-regulated or down-regulated of each group respectively.
A4-detailed analysis at the level of the gene with significant transcription regulation trend was included in an independent manuscript that is still under review at the Radiation Journal. We have alluded to this in the test from those who are interested in learning more about the full picture of this work. We will be glad to share the preprint version of the paper with the reviewer. If interested, we can communicate the manuscript through the Genes Journal team.
Data on body weight change after X-ry irradiation is necessary to infer the condition of the mouse.
A5- These data were tallied and reported in another report. In general, animals showed less than 20% weight loss in animals at euthanasia. The study IACUC approved protocol required euthanasia if animals lost more than 20% of their weight under humane endpoint criteria.
The reason for gene induction or suppression is the direct effect of radiation or the indirect effect.
Please clarify the pathway that was only partially changed is really induced or suppressed by radiation.
A6-Not clear which pathway the reviewer is referring to. We will be glad to clarify if the reviewer can be more specific (page#, Line#).
For example, from Figures 2 and 3, the GP6 signaling pathway remains unchanged at 3 Gy. Why doesn't it change with 3 Gy?
A7- This is a good observation. Responses to radiation in the 1-6Gy radiation range are known to be stochastic. The regulation at 3Gy is another evidence of the nature of these responses.
Also, the Apelin Liver Signaling Pathway changes only at 20 Gy, and it is not possible to determine whether it is due to radiation or just suppressed because it is in a lethal state.
A8- Because all animals in the control group survived the experiment it is fair to assume that the lethal state is due to radiation. Both conditions are induced by radiation.
Reviewer 2 Report
Accept the manuscript.
Author Response
We thank the reviewer for his time and efforts!